# Antibody Response after BNT162b2 Vaccination in Healthcare Workers Previously Exposed and Not Exposed to SARS-CoV-2

**DOI:** 10.3390/jcm10184204

**Published:** 2021-09-17

**Authors:** Marcello Salvaggio, Federica Fusina, Filippo Albani, Maurizio Salvaggio, Rasula Beschi, Emanuela Ferrari, Alberto Costa, Laura Agnoletti, Emanuela Facchi, Giuseppe Natalini

**Affiliations:** 1Department of Anesthesia and Intensive Care, Fondazione Poliambulanza Hospital, 25128 Brescia, BS, Italy; marcello.salvaggio@poliambulanza.it (M.S.); filippo.albani@poliambulanza.it (F.A.); rasula.beschi@poliambulanza.it (R.B.); emanuela.ferrari@poliambulanza.it (E.F.); alberto.costa@poliambulanza.it (A.C.); giuseppe.natalini@poliambulanza.it (G.N.); 2Department of Anesthesia and Intensive Care, Santa Chiara Hospital, 38100 Trento, TN, Italy; Dr.mauriziosalvaggio@gmail.com; 3Department of Laboratory Medicine, Fondazione Richiedei, 25064 Gussago, BS, Italy; Laura.agnoletti@richiedei.it; 4Health Management, Fondazione Richiedei, 25064 Gussago, BS, Italy; Dirsan.gussago@richiedei.it

**Keywords:** SARS-CoV-2, BNT162b2, vaccination, antibody response, COVID-19

## Abstract

The Pfizer/BioNtech Comirnaty vaccine (BNT162b2 mRNA COVID-19) against SARS-CoV-2 is currently in use in Italy. Antibodies to evaluate SARS-CoV-2 infection prior to administration are not routinely tested; therefore, two doses may be administered to asymptomatic previously exposed subjects. The aim of this study is to assess if any difference in antibody concentration between subjects exposed and not exposed to SARS-CoV-2 prior to BNT162b2 was present after the first dose and after the second dose of vaccine. Data were retrospectively collected from the clinical documentation of 337 healthcare workers who underwent SARS-CoV-2 testing before and after BNT162b2. Total anti RBD (receptor-binding domain) antibodies against SARS-CoV-2′s spike protein were measured before and 21 days after the first dose, and 12 days after the second dose of BNT162b2. Twenty-one days after the first dose, there was a statistically significant difference in antibody concentration between the two groups, which was also maintained twelve days after the second dose. In conclusion, antibody response after receiving BNT162b2 is greater in subjects who have been previously exposed to SARS-CoV-2 than in subjects who have not been previously exposed to the virus, both after 21 days after the first dose and after 12 days from the second dose. Antibody levels, 21 days after the first dose, reached a titer considered positive by the test manufacturer in the majority of subjects who have been previously infected with SARS-CoV-2. Evaluating previous infection prior to vaccination in order to give the least effective number of doses should be considered.

## 1. Introduction

At the end of December 2020, the Pfizer/BioNtech Comirnaty vaccine (BNT162b2 mRNA COVID-19) against SARS-CoV-2 was listed for WHO Emergency Use Listing. 

BNT162b2 is an mRNA vaccine, and the vaccination series is made up of two doses which are given 3 weeks apart. In Italy, since 3 March 2021, no vaccination is recommended for the first three months after infection for previously infected individuals who have tested positive for SARS-CoV-2, and after 6 months from infection a single dose may be administered, as per the Ministry of Health Communication (https://www.trovanorme.salute.gov.it/norme/renderNormsanPdf?anno=2021&codLeg=79033&parte=1%20&serie=null, accessed on 15 June 2021). Antibodies are not routinely tested, though, and therefore some asymptomatic subjects might receive two doses, because their previous exposure to SARS-CoV-2 might be undetected. Nonetheless, recent data [1,2] suggest that previously infected individuals have a heightened antibody production compared to subjects who have had no previous exposure to SARS-CoV-2. 

The aim of this study is to assess if any difference in antibody concentration between subjects previously infected with SARS-CoV-2 and subjects not previously infected with SARS-CoV-2 was present, after receiving the first dose and after receiving the second dose of BNT162b2 mRNA COVID-19 vaccine.

## 2. Materials and Methods

Data were retrospectively collected from the clinical documentation of healthcare workers older than 18 years of age, working in Fondazione Richiedei (Gussago, BS, Italy), who underwent SARS-CoV-2 testing before and after BNT162b2 vaccination between 1 January 2021 and 1 April 2021, as planned by the Department of Preventive Medicine of the Foundation. Exclusion criteria were: pregnancy, age < 18 years, not having received two doses of SARS-CoV-2 vaccine. The data were anonymized. The study protocol was approved by Brescia’s Ethical Committee (NP4940). The need for consent was waived due to the retrospective nature of the study.

Blood samples were obtained before the first BNT162b2 dose, 21 days after the first dose and 12 days after the second dose, in order to evaluate the presence or absence of total anti RBD (receptor-binding domain) antibodies against SARS-CoV-2′s spike protein (including IgGs), using Roche COBAS e601 automated immunoassay analyzer. Serum was separated by centrifugation at 3800 rpm for 15 min immediately after collection. According to the manufacturer’s instructions, an anti SARS-CoV-2 antibody concentration of less than 0.8 U/mL was considered negative, while a concentration equal or greater to 0.8 U/mL was considered positive. The maximum detected antibody concentration was 5000 U/mL.

The primary outcome of the study was to assess if any difference in antibody concentration between subjects previously infected with SARS-CoV-2 and subjects not previously infected with SARS-CoV-2 was present, after receiving the first dose and after receiving the second dose of BNT162b2 mRNA COVID-19 vaccine.

A secondary analysis was conducted in order to evaluate any differences in antibody concentration between symptomatic and asymptomatic subjects in the exposed cohort. Subjects were defined as symptomatic in presence of fever (>37.5 °C), ageusia, anosmia, flu-like symptoms (sore throat, cough, runny nose). The timepoints were the same ones used in primary analysis.

### Statistical Analysis

Continuous variables were described with median (1st–3rd quartiles) and analyzed with the Mann–Whitney U test/Wilcoxon rank-sum test in accordance with their non-normal distribution. Sample size calculation was estimated as described by Shieh et al. [3].

For the sample size calculation, we estimated the risk of having been previously infected with SARS-Cov-2 as 30% for healthcare workers. Considering alpha type one error rate set at 0.05, we calculated that data on 300 subjects would be necessary to obtain a power of 0.8, considering as 0.6 the probability of finding a higher logarithm of antibody concentration in exposed subjects compared to non-exposed ones.

The secondary analysis was conducted using Wilcoxon rank-sum test.

Statistical analyses were performed with R (R Core Team, 2020, R Foundation for Statistical Computing, Vienna, Austria) [4].

## 3. Results

Three-hundred and seventy-four subjects were enrolled in the study. Three-hundred and thirty-seven subjects received the two doses of vaccine and were therefore included in the study. Of those, 209 were not previously exposed to SARS-CoV-2 while 128 had detectable antibodies before receiving the first BNT162b2 dose. Exposed subjects were younger than non-exposed subjects, with no difference between males and females between the two groups.

Twenty-one days after the first dose, there was a statistically significant difference in antibody concentration between the two groups. This difference was also maintained twelve days after the second dose (see Table 1).

The natural response in subjects exposed to SARS-CoV-2 is higher than the response induced by the first dose of vaccine in non-exposed subjects (*p* < 0.001), but not higher than the vaccine induced response after the second dose (*p* < 0.001) in exposed subjects, as shown in Figure 1.

The antibody concentration in exposed subjects after the first dose of vaccine was higher than the antibody concentration in non-exposed subjects after the second dose of vaccine (*p* < 0.001).

Fifty-five subjects out of 111 (50%) in the exposed cohort were asymptomatic, while for the remaining 17 patients data were not available. We found no statistically significant difference between asymptomatic and symptomatic subjects at baseline (*p* = 0.59), 21 days after the first dose (*p* = 0.16), and 12 days after second dose (*p* = 0.67). There were no differences in age and sex between the two cohorts (see Figure 2 and Table 2).

## 4. Discussion

This study shows that the antibody response in subjects who have been previously exposed to SARS-CoV-2 is greater than the one in subjects who have not been previously exposed to SARS-CoV-2, both 21 days after the first dose and 12 days after the second dose. Antibody levels, 21 days after the first dose, reached a titer considered positive by the test manufacturer in the majority of subjects who had been previously infected with SARS-CoV-2. No difference in antibody concentration was found between symptomatic and asymptomatic subjects at all tested timepoints.

This confirms the findings made by Gobbi et al. [2] who found that the anti-SARS-CoV-2 IgG RBD response 7 days after the first dose of vaccine was higher in previously exposed subjects than in non-exposed subjects, with non-exposed subjects reaching high titer antibodies only after the second vaccine dose. Moreover, similarly to our data, the antibody concentration in exposed individuals 21 days after the first vaccine dose was not significantly different from those observed in non-exposed subjects 7 days after the second vaccine dose. Differently from our study, Gobbi et al. found that the natural response in exposed subjects was higher than the antibody response 7 days after the first dose of vaccine in non-exposed subjects, but not 21 days after. This could be due to the fact that in our study we measured total anti SARS-CoV-2 RBD antibodies, while Gobbi et al. evaluated IgGs only.

This data are in agreement with other previously published research [5,6,7] showing that a high antibody titer is reached within days after vaccination in previously infected subjects. Of note, this appears to be true not only for the humoral immune response, but also for the cellular immune response [5], supporting the indication to give only a single dose of BNT162b2 to subjects who have been previously infected with SARS-CoV-2, as per Italian Ministry of Health’s Communication (https://www.trovanorme.salute.gov.it/norme/renderNormsanPdf?anno=2021&codLeg=79033&parte=1%20&serie=null, accessed on 3 March 2021). In fact, as discussed by Levi et al. [6], the administration of a second dose in previously infected subjects might lead to antigen exhaustion [8], and therefore be detrimental.

In our cohort, no difference in antibody concentration was found between symptomatic and asymptomatic subjects both before vaccination and after the first and second doses of BNT162b2. This result is particularly interesting because other authors [9,10] found lower levels of both cellular and humoral immunity in asymptomatic individuals following natural infection. Our study is characterized by a larger sample size than other studies [9,10]; moreover, in our analysis, there was no significant age difference between the two cohorts (median age for asymptomatics 44.5 (IQR 38.5–51.5) years vs. 47.5 (IQR 38.0–54.0) for symptomatics, *p* = 0.38), while in Mazzoni et al.’s paper [10], the median age was 29 years for asymptomatics and 85.5 for symptomatics. This might contribute to the difference in results, and further research is needed on the subject.

BNT162b2 contains lipid vesicle nanoparticles carrying synthetic mRNA which encodes the sequence of the coronavirus’s spike protein (S-protein), enabling delivery of the mRNA into host cells in order to allow expression of the SARS-CoV-2 S antigen [11]. An immune response to the S antigen (from the viral spike protein) is then elicited, providing protection against infection [12]. The vaccination series in Italy, at the moment, includes two doses 21 days apart (the second dose can be postponed, but it is recommended that it is administered no later than 42 days after the first). The schedule is the same for all subjects who have not tested for SARS-CoV-2, and previously exposure to SARS-CoV-2 is not routinely evaluated. In light of these data, we believe that the evaluation of previous SARS-CoV-2 infection before starting the vaccination cycle could be useful. Side effects to BNT162b2 have been reported in recent months. The most common ones are pain at the injection site, tiredness, headache, muscle pain, chills, joint pain, and fever [13]. More severe ones, although less frequent, have been reported [1]. These include lymphadenopathy [1], allergic reactions and anaphylaxis [14], deep vein thrombosis [15], Vaccine–Induced Immune Thrombotic Thrombocytopenia (VITT) [16] and, possibly, death. Each vaccine administration is linked to side effects, which are more common in previously infected individuals [13]. Therefore, we believe that evaluating previous infection with SARS-CoV-2 and modifying the vaccination schedule in previously exposed asymptomatic subjects in order to give the least effective number of doses would be of value in order to reduce vaccine related adverse reactions. It would also be important in the case of a vaccination shortage and to reduce costs linked to unnecessary administration. This study can constitute a significant contribution on a subject in which data are still lacking and could be integrated with future research.

This study has some limitations. First, it is a single center study. Second, some of the subjects in the non-exposed group might have been infected with SARS-CoV-2 during the study period. Moreover, further studies on long term follow-up are needed in order to evaluate the duration of a high antibody concentration.

## 5. Conclusions

Antibody response after receiving BNT162b2 is greater in subjects who have been previously exposed to SARS-CoV-2 than in subjects who have not been previously exposed to the virus, both 21 days after the first dose and 12 days after the second dose. Antibody levels, 21 days after the first dose, reached a titer considered positive by the test manufacturer in the majority of subjects who have been previously infected with SARS-CoV-2. Evaluating previous SARS-CoV-2 infection prior to BNT162b2 vaccination in order to give the least effective number of doses should be considered.

## Figures and Tables

**Figure 1 jcm-10-04204-f001:**
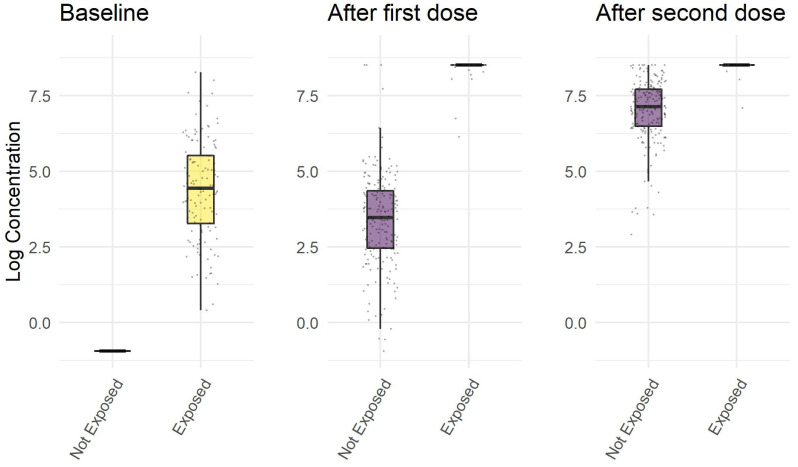
Logarithmic concentration of anti RBD antibodies against SARS-CoV-2′s spike protein (including IgGs) before vaccination with BNT162b2, 21 days after the first dose of BNT162b2, 12 days after the second dose of BNT162b2.

**Figure 2 jcm-10-04204-f002:**
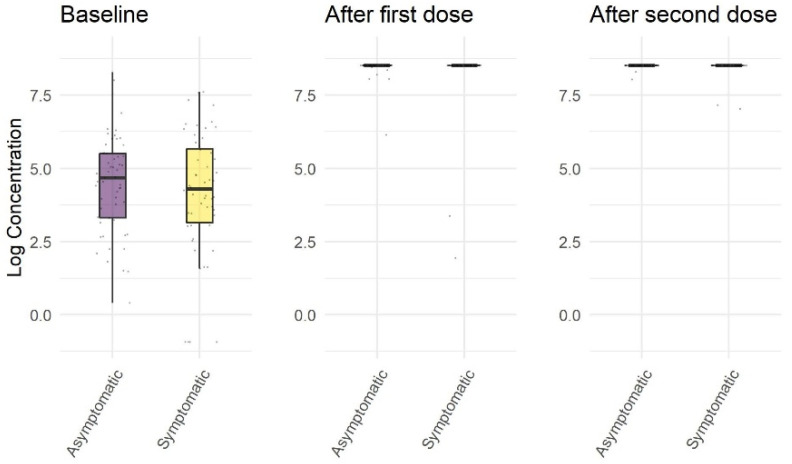
Logarithmic concentration of anti RBD antibodies against SARS-CoV-2′s spike protein (including IgGs) before vaccination with BNT162b2, 21 days after the first dose of BNT162b2, 12 days after the second dose of BNT162b2 in symptomatic and asymptomatic subjects.

**Table 1 jcm-10-04204-t001:** Antibody concentration before and after first and second dose of BNT162b2.

	Non Exposed Subjects	Exposed Subjects	*p*
Ab concentration before first dose, U/mL	0.4 (0.4–0.4)	85 (26–250)	<0.001
Ab concentration 21 days after first dose, U/mL	32 (12–78)	5001 (5001–5001)	<0.001
Ab concentration 12 days after second dose, U/mL	1258 (657–224)	5001 (5001–5001)	<0.001
age, years	50 (42–57)	46 (39–53)	0.003
sex, female	157 (80%)	102 (85%)	0.34

Data are shown as median (1st–3rd quartiles) or count (%). Abbreviations: Ab = anti RBD antibodies against SARS-CoV-2′s spike protein (including IgGs).

**Table 2 jcm-10-04204-t002:** Antibody concentration before and after first and second dose of BNT162b2 in symptomatic and asymptomatic subjects.

	Asymptomatic Subjects	Symptomatic Subjects	*p*
Ab concentration before first dose, U/mL	108.0 (27.3 to 247.9)	73.4 (23.1 to 290.7)	0.59
Ab concentration 21 days after first dose, U/mL	5001.0 (5001.0 to 5001.0)	5001.0 (5001.0 to 5001.0)	0.16
Ab concentration 12 days after second dose, U/mL	5001.0 (5001.0 to 5001.0)	5001.0 (5001.0 to 5001.0)	0.67
age, years	44.0 (38.5 to 51.5)	47.5 (38.0 to 54.0)	0.38
sex, female	47 (85.5)	52 (92.9)	0.34

Data are shown as median (1st–3rd quartiles) or count (%). Abbreviations: Ab = anti RBD antibodies against SARS-CoV-2′s spike protein (including IgGs).

## Data Availability

The datasets are available from the corresponding author on reasonable request.

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
