# Peer review of "Antibody Response after BNT162b2 Vaccination in Healthcare Workers Previously Exposed and Not Exposed to SARS-CoV-2"

_jcm, 2021, doi:10.3390/jcm10184204_

Round 1
Reviewer 1 Report
This study investigates the levels of anti-SARS-CoV-2 antibodies following BNT162b2 vaccination. The authors show that subjects with previous SARS-CoV-2 infection display a significant immune response already after the first dose of the vaccine. These findings have already been reported in the literature by several groups, both at humoral and cellular level. For this reason, the discussion should be implemented with additional references, discussing similarities and differences with other studies (for example doi: 10.1172/JCI149150; doi: 10.1056/NEJMc2101667; doi: 10.1172/JCI149154).
Additional comments:
-Figure1: in x-axis and in the caption it is not clear what 0 and 1 stand for. I guess 0 are unexposed and 1 are exposed subjects.
-In the discussion, authors state "Antibody levels, 21 days after the first dose, have reached a level considered protective against SARS-CoV-2 infection in the majority of subjects who have been previously infected with SARS-CoV-2." Which is the protective level, and how did the authors define it?
-authors should include a clinical characterization of the enrolled exposed subjects. How many were asymptomatic? Are there any differences in the antibody levels following vaccination between subjects with history of symptomatic or asymptomatic infection? This is of importance, as the authors suggest to modify the vaccination schedule also in asymptomatic subjects, but it has previously been shown that asymptomatic individuals have lower levels of both cellular and humoral immunity following natural infection (doi: 10.1038/s41591-020-0965-6; doi: 10.1002/eji.202048915).
Reviewer 2 Report
Overall a well designed and executed study emphasizing an observation which has been previously shown.
Author Response
Thank you for taking the time to review our paper and for your comments.
Round 2
Reviewer 1 Report
I would like to thank the authors for addressing my concerns. Regarding the Figure and Table related to differences between symptomatic and asymptomatic subjects, I suggest including in the manuscript. Indeed, they provide important findings, confirming that one vaccine dose is equally effective in previously infected subjects, irrespective of disease severity.